# Heterogeneity in Mechanical Properties of Plant Cell Walls

**DOI:** 10.3390/plants13243561

**Published:** 2024-12-20

**Authors:** He Zhang, Liang Xiao, Siying Qin, Zheng Kuang, Miaomiao Wan, Zhan Li, Lei Li

**Affiliations:** 1School of Advanced Agricultural Sciences, Peking University, Beijing 100871, China; xiaoliang@pku.edu.cn (L.X.); wanmiaomiao@pku.edu.cn (M.W.); 2School of Life Sciences, Peking University, Beijing 100871, China; qinsiying@pku.edu.cn (S.Q.); 2001110464@stu.pku.edu.cn (Z.L.); 3Institute of Biotechnology, Beijing Academy of Agriculture and Forestry Sciences, Beijing 100097, China; kuangzheng@pku.edu.cn; 4Shandong Laboratory of Advanced Agricultural Sciences at Weifang, Peking University Institute of Advanced Agricultural Sciences, Weifang 261000, China

**Keywords:** cell wall, mechanical properties, heterogeneity, atomic force microscopy

## Abstract

The acquisition and utilization of cell walls have fundamentally shaped the plant lifestyle. While the walls provide mechanical strength and enable plants to grow and occupy a three-dimensional space, successful sessile life also requires the walls to undergo dynamic modifications to accommodate size and shape changes accurately. Plant cell walls exhibit substantial mechanical heterogeneity due to the diverse polysaccharide composition and different development stages. Here, we review recent research advances, both methodological and experimental, that shed new light on the architecture of cell walls, with a focus on the mechanical heterogeneity of plant cell walls. Facilitated by advanced techniques and tools, especially atomic force microscopy (AFM), research efforts over the last decade have contributed to impressive progress in our understanding of how mechanical properties are associated with cell growth. In particular, the pivotal importance of pectin, the most complex wall polysaccharide, in wall mechanics is rapidly emerging. Pectin is regarded as an important determinant for establishing anisotropic growth patterns of elongating cells. Altogether, the diversity of plant cell walls can lead to heterogeneity in the mechanical properties, which will help to reveal how mechanical factors regulate plant cell growth and organ morphogenesis.

## 1. Introduction

Plants arise from a fixed location and grow to occupy a three-dimensional space [1]. This requires the plant structures to be flexible to allow appropriate growth as well as rigid enough to support the increasing body mass and be resilient to harsh environments. As a result, plants have developed unique cell walls as one of their ingenious strategies to thrive on Earth [2]. Plant cell walls consist of a complex polysaccharide-rich network and display marked diversity in different species, developmental stages, and even cell types [3]. For example, the primary cell walls and the secondary cell walls both comprise cellulose and hemicellulose. However, there is a large amount of pectin in the primary cell walls, while the secondary cell walls have a large proportion of lignin. In addition, the primary cell walls of dicotyledonous plants usually contain mostly pectin, while those of monocotyledonous plants are comprised mostly of hemicellulose [4]

Plant growth starts from meristematic cells, which undergo cell division, expansion, differentiation, and patterning to develop specialized organs and attain the organism’s size [1,5]. Cell expansion is a pivotal determinant for the final shape and size of the plant organs. Most cells may enlarge >1000-fold in volume to morph into a variety of shapes for plant organs [6]. Plants consist of about 35 cell types with distinctive positions, shapes, final sizes, and wall characteristics [7]. For example, as shown in Figure 1, the pavement cells in the leaf epidermis of a dicotyledonous plant (*Arabidopsis*) are multi-lobed like jigsaw puzzle pieces [8], while the pavement cells of monocotyledons (maize) are parallel and arranged in rows with weak undulations [9]. Even the identical cell type of the same plant is distinguished from others. The epidermal cells of the hypocotyl are tightly arranged squares while those of the petal are conical [10,11]. Meanwhile, some structures exhibit certain similarities, such as a polygonal-net-like structure of the surface of the pollen grain as well as the cross-section in the young stems of the poplar (Figure 1) [12,13]. Extensive research on cell expansion has shown that expansive growth is a mechanical process, which balances the internal turgor pressure and the external stresses from the cell walls [14,15]. Thus, the growth of cells enveloped by cell walls requires integrating the cell wall network via the regulation of cell wall synthesis, degradation, and modifications and interaction with the membrane proteins [16,17]. The breaking of the equilibrium caused by cellular growth leads to heterogeneity in the mechanical properties of cell walls, which can be detected by nanoindentation techniques such as atomic force microscopy (AFM) [18,19].

Here, we initially give a brief overview of the main cell wall components, especially pectin, and recent technologies for detecting the mechanical properties of cell walls. Plant organ morphogenesis is determined by the extensibility of the cell wall, which is controlled dynamically by the organization and modification of its components, resulting in changes in the cell wall’s mechanical properties. Thus, we summarize the widespread application of AFM in detecting the mechanical properties of plant cell walls and discuss the concepts regarding the significance of heterogeneity in the mechanical properties of plant cell walls for plant growth and development.

## 2. Cell Wall Components and Dynamics

There are three different basic categories of cell walls in vascular plants, namely, the primary wall, the secondary wall, and the middle lamella (Figure 2). These three types of walls are different from each other not only in chemical composition but also in their respective functions. In growing plant organs, virtually all cells have the middle lamella and the primary cell wall. The middle lamella is a pectin layer that cements the two adjoining primary cell walls together (Figure 2) [20,21,22]. Plant cells go through apparent phases of development. In these phases, primary cell walls are the major physical and structural determinants for the rate and direction of cell expansion [23,24]. The primary cell walls of higher plants include three major types of polysaccharides: cellulose, hemicellulose, and pectin (Figure 2). These polysaccharides form a dynamic and complex array in the structure of cell walls [25,26,27]. A typical wall is made of a crystalline backbone composed of cellulose microfibrils interconnected by hemicelluloses and embedded in a pectin matrix together with a number of structural proteins [28,29].

Cellulose is a linear, unbranched polymer composed of β-1,4-glucose units, which are linked into a flexible network of other matrix polysaccharides [1,30,31]. In current models, cellulose presents in the form of crystalline cellulose microfibrils which are the major load-bearing components of the cell wall. Each neighboring glucose residue rotates 180° toward the chain axis to form a set of parallel chains which are subsequently stacked into crystalline cellulose microfibrils. The degree of polymerization is a parameter of cellulose defined by the molecular weight and length of the microfibrils. Cellulose is synthesized by cellulose synthase complexes that are assembled in the Golgi apparatus and then delivered to the plasma membrane [32,33,34]. Cellulose synthase complexes define the orientation for producing the cellulose microfibrils, which travel along cortical microtubule paths [32]. Hemicelluloses are present in all terrestrial plants and these polysaccharide chains consist of xyloglucans, xylans, mannans, and glucomannans [35]. Xyloglucans are the main group of hemicelluloses with a branched structure comprising a β-1,4-linked glucan backbone and side chains including xylose, galactose, and fucose residues [36]. Given the diverse structures, biochemical properties, and cellular distribution, many enzymes and pathways have been identified as contributors to the biosynthesis of hemicellulose constituents [37,38,39,40,41]. The function of hemicelluloses is reported to influence the cell wall mechanics by acting as an adhesive layer among cellulose microfibrils, which contributes to strengthening the cell wall and controlling the movement of microfibrils during the growth process in response to external forces [42,43,44,45,46].

Pectin belongs to a unique class of cell wall components that are distinguished from other matrix polysaccharides by possessing a large number of acidic sugars as galacturonic acid and a smaller amount of glucuronic acid [47,48,49]. Pectin biosynthesis also starts at the Golgi apparatus, which is a multi-compartment organelle and cell wall polysaccharide synthesis “factory”. Recent advances have revealed some parts of the machinery for pectin biosynthesis, but plentiful mysteries remain [47,50,51]. As a measure of the complexity of pectin biosynthesis, at least 67 different types of enzymatic activities are believed to be required [52]. It has been reported that there are over 730 genes encoding putative glycosyltransferases or glycosyl hydrolases in the *Arabidopsis* genome [25]. Proteomics analyses of isolated Golgi apparatus have identified additional putative proteins or enzymes [53]. Because most of these enzymes are likely integral membrane proteins in their active form and there is a lack of robust in vitro assays to validate their function, it is difficult to isolate and track a single specific gene involved in pectin synthesis or modification.

There are three types of pectin polysaccharides, namely, homogalacturonan (HG), rhamnogalacturonan I (RG-I), and rhamnogalacturonan II (RG-II). The primary walls in the *Arabidopsis* hypocotyl are composed of up to 35% pectin [28]. Of these, HG is the predominant form and constitutes about two-thirds of all cell wall pectin [54]. The chemical structure of HG is an unbranched chain of α-1,4-linked D-galacturonic acid subunits, which can be chemically modified [47,48,49]. The addition of O-acetyl ester groups on the C2 and C3 atoms of galacturonic acid subunits is called pectin acetylation. Pectin methylesterification refers to the addition of methyl carboxyester groups on C6 atoms. As the second most abundant polymer of pectin, RG-I takes about 20% to 35% of total pectin, which comprises alternating galacturonic acid and rhamnose residues. In addition, most of the rhamnose residues are substituted with sugar sidechains such as galactan, arabinan, and arabinogalactans [16,55]. RG-I is recognized as the most structurally heterogeneous of the pectic polymers. There is some variation in the composition and prevalence of the RG-I sidechains between different cell types and species [49]. RG-II is a group of highly conserved and complex pectic polymers representing only about 10% of total pectin. RG-II exists in the form of a backbone of HG decorated with various side chains including more than 20 different glycosyl linkages and 13 diverse sugars [16]. The sidechain compositions are distinct at different developmental stages or among different species [49].

The accurate arrangement of pectin domains in the cell wall is mostly unclear, but there is a mix of covalent and non-covalent interactions combining pectin to a crosslinked matrix. For instance, RG-II dimerizes by forming borate diester bonds between two RG-II molecules. There is a noncovalent calcium-mediated interaction between demethyl-esterified portions of HG, known as “egg-box” structures, which play an important role in some biological processes [56]. Pectin has also been reported to interact with cellulose by galactan or arabinan side chains of RG-I in vitro [57]. More interestingly, the polysaccharides of cell walls undergo liquid–liquid phase separation in pollen grains, which broadens the understanding of the mechanism for cell wall matrix patterning [12].

Pectin is a major determinant of the biophysical properties of the cell wall such as adhesion, cohesion, and extensibility. Recent studies have uncovered previously unappreciated roles of pectin as a key factor in wall structure, cellular growth, cell-to-cell communication, and organ morphogenesis [47,58]. Mutations altering the pectin content result in significant changes in plant development and organ growth, such as *gaut8*, *gaut12*, and *gatl1* mutants with severe dwarfed growth and sterility [55]. The ablating or overexpressing of *POLYGALACTURONASE INVOLVED IN EXPANSION3* (*PGX3*) affects the cotyledon shape and stomata development [56]. Mutations in *Arabidopsis RHAMNOSE BIOSYNTHESIS 1* (*RHM1*), which is required for the synthesis of RG-I, cause significant left-handed helical growth of the petals and roots [59]. Besides the synthesis and degradation of pectin, plants regulate their growth through the modification of pectin in changing the degree of methyl esterification or the acetylation level. For example, the HG methyltransferase QUASIMODO2 mutation (*qua2*) reduced cell adhesion in the cotyledon and hypocotyl [60]. Pectin de-methylesterification of longitudinal cell walls is one of the events underlying cell wall loosening during the elongation of dark-grown hypocotyls [61,62].

## 3. Techniques and Tools for Probing Biophysical and Biochemical Properties of Cell Walls

During plant development, cells might enlarge extremely compared with their original sizes [63]. Turgor pressure, also called hydrostatic pressure, is the pressure exerted by the osmotic flow of water and provides the force that pushes the plasma membrane against the cell wall. The expansion of plant cells is primarily determined by the force balance between the turgor and the tensile resistance of the cell walls. The cell growth process can cause changes in the physicochemical status inside and outside the cell. This has inspired biologists to use various ingenious methods to noninvasively measure the mechanical and chemical properties and dynamics over the last half century [64,65,66,67,68]. Concurrently, multiple imaging techniques have been developed to illustrate the surface structure and biochemical properties of the cell wall, which have added straightforward crucial information about cell wall development and function [69,70]. These imaging methods have two categories, namely, labeling-based and label-free methods. Labeling-based techniques include histochemical/cytochemical techniques, immunolabeling techniques, and genetically encoded reporter techniques [71,72,73,74]. Cellulose can be detected using some dyes, such as a fluorescent brightener FB28, as shown in the pavement cells of the maize leaf (Figure 1) [75]. Pontamine Fast Scarlet 4B is another widely used fluorescent dye to stain cellulose and is especially suitable for confocal microscopy [76]. In addition, there is a series of monoclonal antibodies for the analysis of cell wall carbohydrates, such as LM19 for unesterified HG and LM20 for high methylester HG. These immunolabeling experiments have been widely used in different tissues and species including *Arabidopsis* leaves, guard cells, seed mucilage, rice pollen cells, and *Vaccinium macrocarpon* pith cells [77,78,79,80,81]. Overall, fluorescence microscopy is becoming an essential tool, and these fluorescent molecules with specific binding affinity to various cell wall polysaccharides are used to visualize and locate cell wall components in plant development and cell biology [82].

Label-free imaging techniques comprise transitioned tools from ultraviolet microscopy, scanning electron microscopy (SEM), and transmission electron microscopy to more advanced techniques such as Fourier transform infrared microspectroscopy, confocal Raman microspectroscopy, coherent anti-Stokes Raman scattering microscopy (CARS), stimulated Raman scattering microscopy (SRS), and AFM [69,83,84,85,86]. Those technologies can generally be divided into two categories: one is based on the chemical bond features and the other is dependent on the physical interactions. Raman scattering microscopy is designed to collect the structure characterization and the chemical identification of plant cell walls based on the inelastic scattering phenomenon, which becomes a powerful approach to capture abundant details with micrometer resolution [87,88]. CARS and SRS have been developed to improve the sensitivity and accuracy of Raman spectroscopy [89]. Both CARS and SRS are successfully applied to visualize the major chemicals and structural composition in *Arabidopsis* and woody plants [77,89]. Some techniques have also been applied to investigate the structure and assembly of cell wall components, including the crystallite shape and size and crystallinity of cellulose, such as X-ray scattering, which reveals the size and orientation of cellulose microfibrils, sum frequency generation spectroscopy, which indicates the meso-scale ordering and crystallinity of microfibrils, and nuclear magnetic resonance, which identifies the structure of individual cell wall polymers [90]. Specifically, imaging techniques, like SEM, provide information on the superficial changes in the cell wall textures and structures (Figure 1).

The other tools based on physical interaction detect mechanical properties and surface morphology simultaneously. Modern methods for probing the mechanical properties of the cell wall, including elasticity, plasticity, viscoelasticity, and viscosity, generally combine micro-indentation techniques with force transducers and rheometry [91]. The principle of micro-indentation is the application of a probe to press the surface of the biological samples whereby the depth or area of the produced indentation and the initial force are used to deduce the mechanical properties of the tested samples [92]. Because the cell wall thickness usually ranges between 0.1 and a few micrometers, it is challenging to use micro-indentation methods in vivo. Within the last decade, nanoindentation methods, such as AFM (Figure 3), have been increasingly applied to analyze cell wall biophysics [93,94,95,96,97]. The principle of AFM is to apply a probe hanging under the cantilever to touch the samples and then collect force-indentation curves at various cell surface points (Figure 3). The curves enable the calculation of the mechanical properties such as the adhesion and elasticity to re-construct the three-dimensional morphology of samples by analyzing the depth data as well [84,98]. The application of AFM has enabled researchers to obtain unprecedented information about the surface architecture, mechanics, and chemical properties of living plants at single-cell or even subcellular levels (Figure 4) [99,100].

## 4. Advanced Applications of AFM in Plant Cell Walls

Owing to technical advancements, a great number of delicate studies have been carried out on multiple species during almost all development stages with the application of AFM [101]. Plant organs usually initiate from the meristem and, for example, the leaf primordia commence from the margin of the shoot apical meristem. Qi et al. found that leaf dorsoventral polarity signaling resulted in mechanical heterogeneity of the cell wall in tomato and *Arabidopsis* by imaging the pectin methyl-esterification distribution and applying AFM to map the wall rigidness [102]. The elastic modulus of the epidermal cell wall in living tomato leaf primordia ranged from 1 to 10 MPa in different leaf domains. As the leaves grow and enlarge, the stiffness changes. Zhang et al. recorded an elastic modulus of about 50 MPa in the epidermal cell wall of mature cotyledons. They illustrated that the leaves were larger than the wild type in mutants with a low pectin level, which allows a softer cell wall, and the same phenotypes were observed in the petals with a modulus of around 200 MPa [77]. The research object of AFM is not limited to dicotyledonous plants but is also used in monocotyledonous plants, such as rice, maize, and wheat [103,104]. For instance, Wang et al. measured the mechanical properties of the cell walls from lodicule cells to reveal a model showing that pectin methylesterification controlled the diurnal flower-opening times in rice [105]. The surface elastic modulus of the lodicules was around 10 MPa in the control lines while that of the mutants was up to 25 MPa or lower than 10 MPa. With the development of operational methods and a variety of working modes, the detection range for AFM is no longer limited to the surface structure of plant tissues. For example, the cellulose microfibrils are not usually exposed on the surface of the cell wall, while Du et al. used AFM to detect cellulose microfibrils from resuspended cell wall samples that were ground in liquid nitrogen and rinsed in buffers [60]. It is now possible to investigate the mechanical properties of internal cells assisted via tissue slicing using AFM, such as research on maize stover stems, maize roots, and poplar stem cells (Figure 4) [106,107,108].

To address the fast-growing changes in certain biological processes, researchers employed simple tissues, such as onion epidermis and pollen grains, to observe the alterations of cell walls (Figure 2) [109,110]. They monitored real-time changes in cell-free strips of onion epidermal walls with different enzyme treatments to assess which polysaccharides bear the mechanical forces of the cell wall by AFM. Their observations showed that the removal of cellulose microfibrils in superficial lamellae softened the wall by reducing the mechanical stiffness but yet did not induce wall loosening. However, HG removal increased the indentation compliance but not the tensile compliance. These findings showed that the arrays of cellulose-hemicelluloses were embedded in the pectin matrix, which acted as “glue” to strengthen the cell wall and allowed the plant cell wall to extend [47,96]. The increasingly quantitative information opens the possibility for mathematical modeling and computational simulation to recapitulate how cell wall elasticity, plasticity, and time-dependent extension are related to dynamic cell wall structures [29,111,112,113,114].

## 5. Heterogeneity in Mechanical Properties of Plant Cell Walls

Although plenty of molecular players have been identified and many conceptual and mathematical models have been proposed, a crystal-clear understanding of how cell wall remodeling mechanistically dictates specific shape change is still pending [15,115,116]. It has been generally accepted that turgor pressure generates an isotropic force on the cell wall. When cells need to achieve directional growth, the chemical components and structure of load-bearing walls are modified to be mechanically anisotropic [14]. The local stiffness of the cell wall is likely to be a result of growth regulation in the development process. Here, we take the hypocotyl elongation and pavement wave pattern as two examples to further illustrate the heterogeneity of plant cell walls.

In darkness, rapid hypocotyl elongation depends on the ability of cells to shift from isotropic to anisotropic growth. This growth symmetry breaking reflects changes in the extensibility of the cell walls. The classical view of the direction of turgor-driven cell expansion is that the cortical microtubule mediates the orientation of cellulose microfibrils. Anisotropic elongation of the hypocotyl is thus canonically attributed to cell wall anisotropy produced by the oriented cellulose fibers. In dark-grown hypocotyl cells, microtubule arrays were indeed observed to be transverse to the longitudinal axis of the cell [117]. After exposure to light, the hypocotyl cells ceased axial extension and the microtubules were rapidly rearranged to be parallel to the growing axis. Using AFM to track the changes in cell wall mechanics in the hypocotyl epidermis, Peaucelle et al. found that an asymmetric loosening of longitudinal anticlinal walls took place without cortical microtubule reorientation prior to growth symmetry breaking [62]. The quantification results showed that the moduli of the transverse walls were 0.2 MPa and those of the longitudinal walls were less than 0.1 MPa in the wild type. However, the moduli of the transverse and longitudinal walls both increased to 0.3 MPa or above in the *PMEI3* overexpression lines and the increments of the longitudinal walls were larger than those of the transverse walls. On the contrary, the moduli of the transverse and longitudinal walls simultaneously fell below 0.1 MPa in the *PME5* overexpression lines. Hence, they interpreted that wall loosening to be triggered by the selective de-methylesterification of pectin in the longitudinal walls where pectin methylesterase activity showed a bipolar distribution. Moreover, they revealed that the subsequent orientation of the microtubule was not required for growth symmetry breaking but contributed to consolidating the growth axis [62]. By carefully monitoring the periodic diurnal variation in hypocotyl growth, Ivakov et al. confirmed that light conditions indeed caused heterogeneity in the cell walls [118]. They further illustrated that cellulose synthesis and cell expansion could be uncoupled and, therefore, were regulated by different mechanisms [118]. Bou Daher et al. quantitatively showed that pectin chemistry and wall elasticity were asymmetric in the etiolated hypocotyl epidermis starting at germination [61]. They concluded that the differential cell wall stiffness, which resulted from the different pectin chemical modifications, triggered growth symmetry breaking followed by cortical microtubule reorientation.

Similarly, localized changes to the cell wall polymers led to heterogeneous biomechanical forces, which were found in the epidermal pavement cells of leaves. The interdigitated jigsaw puzzle shape of pavement cells has been observed in many species, such as in the cotyledon and the true leaves in *Arabidopsis* as well as maize (Figure 1). A widely accepted model for the internal mechanism of pavement cell shape and undulation formation was represented by the co-interactions of the ROP signaling and phytohormone signaling pathways [9]. However, the precise function of cell wall components has been a hot topic of debate for a long time. Advances in biophysical measurements and quantitative imaging technologies have resulted in some theories to explain the cell wall biology impact on pavement cells to develop undulating perimeters [119]. Majda et al. used computational modeling to show that the pavement cell shapes within an epidermis must be related to the mechanical heterogeneities across and along the anticlinal cell walls between adjacent cells under tension [95]. They scanned 16 mutants deficient in cell wall components to measure the cell geometry and demonstrated that wavy cell contours relied on the cell wall composition. Moreover, they detected the polar distribution of pectin and heterogeneity in the mechanical properties along cell walls by performing immunocytochemistry via TEM and AFM. In addition, Altartouri et al. used the anisotropy1 mutant (*any1*), which maintained the overall cellulose content but reduced the cellulose crystallinity, to demonstrate that the initiation of undulations is related to the local enrichment in demethylated pectin, and, subsequently, the deposition of cellulose microfibrils promoted the lobe development in pavement cells [120]. Altogether, in the epidermal pavement cells of leaves, the local mechanical heterogeneity in the cell walls contributed to the morphogenesis of wavy cell shapes [121].

Heterogeneity exists not only in leaves and hypocotyls but also in various cell types of plants. Here, we displayed Young’s modulus (i.e., a measure of the elasticity) from six different plant organs by applying AFM (Figure 4). Under our experimental conditions, the modulus of the growing pollen tube in *Nicotiana tabacum* was the smallest at around 5 Mpa, followed by the xylem tissue from the young poplar stem and the cotyledon of *Arabidopsis* with moduli of about 10 and 20 Mpa, respectively. However, in the mature tissues, the stiffness of the cell walls was higher. The maximum moduli on the pavement cells of the maize leaf can reach 50 MPa, while those on the epidermal cells of the *Arabidopsis* petals are ten times higher, up to 500 MPa. The outer shell of the pollen grain, namely the exine, is recognized as one of the most rigid substances in nature [122]. The moduli of the exine were up to 3 GPa, which is over 600 times higher than that of the pollen tubes (Figure 4). These results are consistent with previous reports [77,107,123,124]. Overall, every cell type exhibits heterogeneity in the cell wall mechanics, and the modulus is different even in the same area of the walls (Figure 4). Thus, mechanical heterogeneity plays a vital role in shaping cell growth and formatting organ development patterns [125].

The underlying molecular mechanism of the mechanical heterogeneities of cell walls still remains unknown. First, it is elusive to determine the essence and chemical modification status of the polymers generating mechanical heterogeneity. Increasing evidence points to heterogeneity in the polysaccharide constituents since the earliest analyses of isolated wall components by chemists [126,127]. In addition, for a long time, chemical isolation has been one of the most common ways of investigating cell wall polysaccharides [127,128]. Although differences in the average wall composition are observed using chemical isolation methods, a good deal of heterogeneity from different cell types or the chemical status could be disguised [129]. Fortunately, a series of advanced in situ methods have been applied to partially make up for this deficiency, such as specific monoclonal antibodies, Raman spectroscopy, AFM (Figure 3 and Figure 4), and so on [130,131,132]. In addition, there is feedback regulation of mechanical heterogeneity in morphogenesis. The dynamic cell wall network produces such local stiffness, which leads to further changes in the cell status, combining the growth signals inside and outside the cells simultaneously. Hamant and his colleague summarized that many development processes rely on the indirect perception of mechanical forces by mechanotransduction pathways [133,134]. For instance, calcium, binding with the demethylesterified pectin of cell walls, could be released and cause conformational changes in downstream factors (like FERONIA) to regulate morphogenesis. In this case, the cell wall components acted as signals to convert the mechanical properties into biochemical cues [17,135]. Therefore, the mechanical properties and the cell wall network are two sets of dynamic, heterogenous, and coordinated regulatory mechanisms in plant development.

## 6. Conclusions and Perspective

Plant cell walls are heterogeneous and dynamic structures because their composition and modification status vary according to development stages and environments. Reflecting this complex nature, about 10% of plant genes are involved in the biosynthesis and regulation of cell walls [46]. Here, we have reviewed methodological and experimental research in recent years that sheds new light on the mechanical heterogeneity of plant cell walls. In particular, with quantitative analysis and noninvasive in vivo imaging, the pivotal importance of pectin in wall mechanics is rapidly emerging. On the one hand, a more predominant role for the pectin network in enabling wall–turgor pressure dynamics is taking shape [136]. On the other hand, pectin is now appreciated as an important determinant for establishing the anisotropic growth patterns of elongating cells [61,62]. As the scientific community continuously makes discoveries, new exciting questions will arise, and old ones will be revisited regarding the relationship between the cell wall and developmental patterning in plants. Given the complexity of cell wall polysaccharides, it is crucial to use multiple experimental and computational techniques for isolating target proteins, developing enzymatic assays, and constructing predictive models. We believe technological development together with the implementation of interdisciplinary approaches will inspire and advance our future understanding in this field.

## Figures and Tables

**Figure 1 plants-13-03561-f001:**
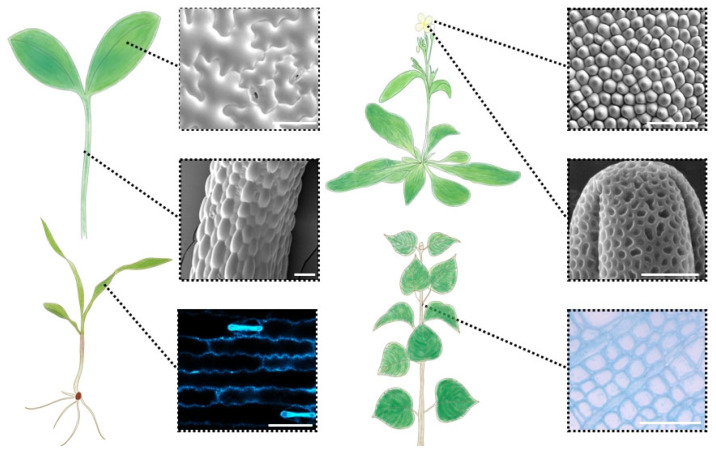
The structures of the diverse cell types of plants. The drawings represent four kinds of plants: seedlings of *Arabidopsis*, maize, and poplar and the adult *Arabidopsis*. The typical cell types in the corresponding area are shown in the dashed boxes. The images of the epidermal cells of the cotyledon, hypocotyl, and petal of *Arabidopsis* as well as the pollen grain were captured by SEM from different plants. The pavement cells of the maize leaf were stained with FB28 to visualize cellulose. The stem cross-section of the young poplar was stained with toluidine blue O staining. The bar of the pollen grain represents 5 μm, and the other bars represent 50 μm.

**Figure 2 plants-13-03561-f002:**
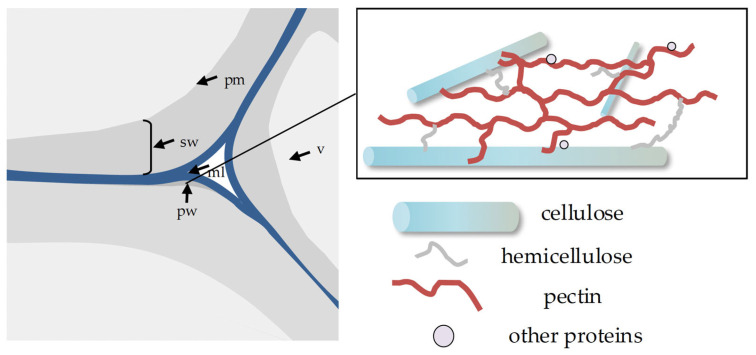
The major cell wall types and the primary cell wall structures. The drawings (**left**) represent the junction of three plant cells. pm, plasma membrane; ml, middle lamella; pw, primary cell wall; sw, secondary cell wall; v, vacuole. The annotation box (**right**) displays the structures of the primary cell wall.

**Figure 3 plants-13-03561-f003:**
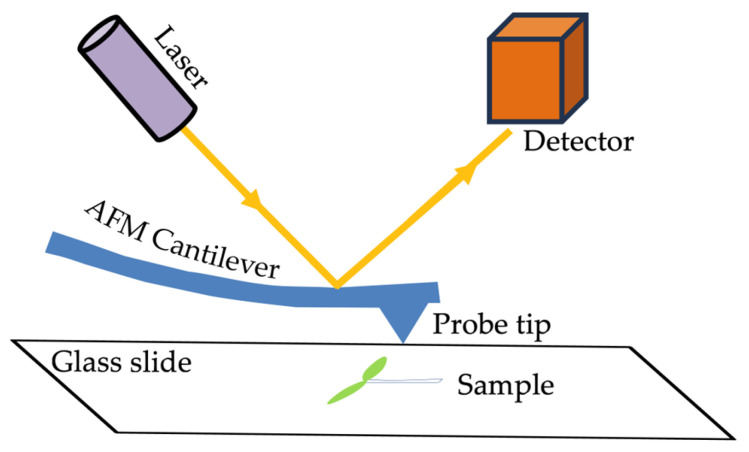
Schematic representation of the AFM. The sample (represented as a young seedling) is placed underneath a probe hanging in a flexible cantilever at a given force. The actual indentation depth is detected by the deformation of the cantilever via a laser deflection. Then, the force curves are captured by the detection unit.

**Figure 4 plants-13-03561-f004:**
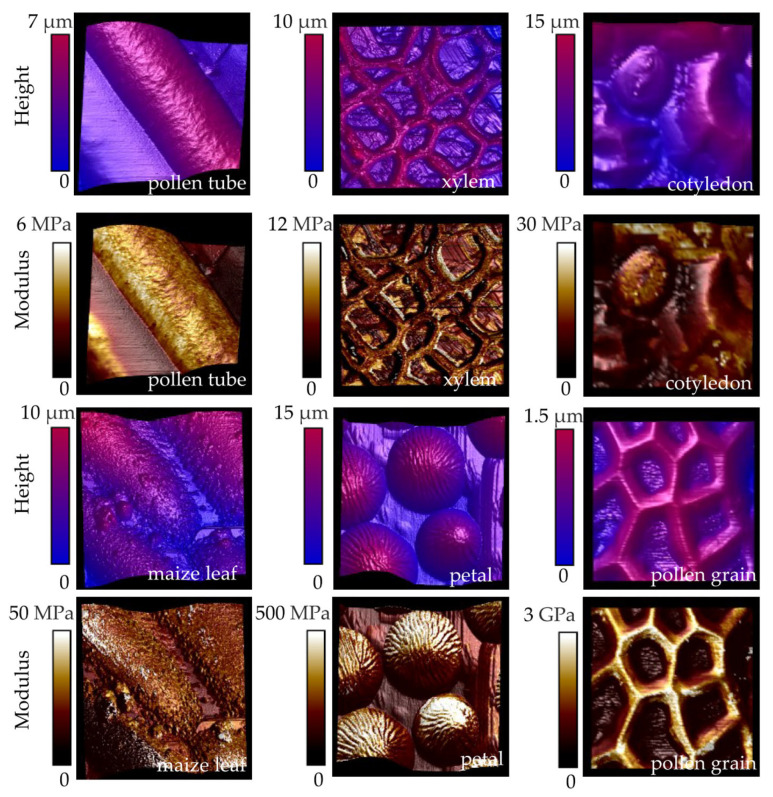
AFM mapping of three-dimensional topography and elastic modulus of epidermal cells from six plant tissues with peak force tapping mode. The distal region of a growing *Nicotiana tabacum* pollen tube, the cross-section of a young poplar stem, the cotyledon of *Arabidopsis* seedlings, a maize leaf, *Arabidopsis* petals, and pollen grains were analyzed. In the upper channel, the colors of the three-dimensional topography represent the distance from the base, which is the deepest point the probe reaches. The cell topography overlaid with the elastic modulus is shown in the bottom channel with the colors indicating elasticity.

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
