# Peer review of "Heterogeneity in Mechanical Properties of Plant Cell Walls"

_plants, 2024, doi:10.3390/plants13243561_

Round 1
Reviewer 1 Report
Comments and Suggestions for Authors
The authors provide a well written and referenced review of the biophysical and biochemical architecture of plant cell walls that comprise various plant structures, emphasizing the underlying biochemical constituents. The diverse structures and associated functional properties of plant cell walls are reviewed primarily by reference to their biochemical heterogeneity. The review serves as a nice background for readers looking to become familiar with the current state of cell wall physical probes and cell morphology for plant cellular structures.
While the review title references mechanical properties that are typically understood to have quantitative descriptions such as stress/strain relationships, elasticity, moduli, and other well-known mechanical parameters, there is not an emphasis connecting morphological information or AFM data with quantitative descriptors of mechanical properties. The references are cited but not elaborated. The review is of less interest, perhaps, to engineering researchers in the field.
I would suggest the authors consider the wording in the title - Mechanical Properties Heterogeneities sounds awkward. Perhaps Mechanical Property Heterogeneities...
Author Response
The authors provide a well written and referenced review of the biophysical and biochemical architecture of plant cell walls that comprise various plant structures, emphasizing the underlying biochemical constituents. The diverse structures and associated functional properties of plant cell walls are reviewed primarily by reference to their biochemical heterogeneity. The review serves as a nice background for readers looking to become familiar with the current state of cell wall physical probes and cell morphology for plant cellular structures.
Reply: Thank you for the encouraging assessment of our work.
While the review title references mechanical properties that are typically understood to have quantitative descriptions such as stress/strain relationships, elasticity, moduli, and other well-known mechanical parameters, there is not an emphasis connecting morphological information or AFM data with quantitative descriptors of mechanical properties. The references are cited but not elaborated. The review is of less interest, perhaps, to engineering researchers in the field.
Reply: Thank you for this insightful suggestion. We have added the content about morphological information or AFM data with quantitative descriptors of mechanical properties in the revised manuscript (line 249 to line 260 and line 302 to line 308).
I would suggest the authors consider the wording in the title - Mechanical Properties Heterogeneities sounds awkward. Perhaps Mechanical Property Heterogeneities...
Reply: Thank you for the advice. We have revised the title as “Heterogeneity in mechanical properties of plant cell walls”.
Reviewer 2 Report
Comments and Suggestions for Authors
This review article provides insights into recent advancements in understanding the heterogeneity in mechanical properties of plant cell wall and its possible reasons such as chemical composition. The application of AFM in unraveling complex plant biological research problems is clearly illustrated in this review. Although the content is very interesting and scientifically significant for publication, the present version requires following minor revisions.
Title
1. ‘Heterogeneity in mechanical properties of plant cell walls’
Introduction
2. Page 1, line 34. Sentence is difficult to follow. ….flexible enough to allow appropriate growth, on the other hand rigid enough to support….
3. Page 1, line 39. Rewrite the sentence on chemical composition of primary and secondary cell wall. Give focus on similarities (cellulose, hemicellulose) and contrasts (pectin and lignin).
4. Page 1, line 43. Remove ‘dividing’
5. Page 2, line 84. ‘embedded in pectic matrix’…
6. Page 2, line 87. Correct as ‘more recent’….
7. Page 2, line 89. Give a brief description on monomer chemistry and organization as polymer with degree of polymerization of cellulose.
8. Page 3, line 138. Remove ‘And’. Pectin has also been reported….
9. Page 3, line 146. Correct as ‘significant change…’
10. Page 4, Figure 1. The SEM figures of pollen grains. Are they from same plant or different? Staining used for cross section of stem. If Safranin is used, it should stain lignified cell wall as red and fast green stain non-lignified cell wall part as green. The provided figure looks like toluidine blue O staining. Please verify the method and provide correct information.
11. Page 5, line 224. AFM method in which mode ‘contact or non-contact mode during tapping?
12. Page 5, line 226. Delete the repeating word ‘owing’
13. Page 6, line 243. It is now possible to investigate ….
14. Page 6, line 251. ..the wall by reducing mechanical stiffness but yet did not…
15. Page 6, line 274. Cellulose fibres or cellulose microfibrils?
16. Page 8, line 348. Figure 2 does not provide immunolabelling figures. Adjust the ‘figure 2’ indication in the correct place. Perhaps as ‘AFM (Figure 2)
Conclusion
17. Page 9, line 361. Correct as ‘modification status’
Comments on the Quality of English LanguageThe quality of English could be improved in many places. For example, use of 'on the other hand' 'and' etc.
Author Response
This review article provides insights into recent advancements in understanding the heterogeneity in mechanical properties of plant cell wall and its possible reasons such as chemical composition. The application of AFM in unraveling complex plant biological research problems is clearly illustrated in this review. Although the content is very interesting and scientifically significant for publication, the present version requires following minor revisions.
Reply: Thank you for the superb comments of our work.
Title
- ‘Heterogeneity in mechanical properties of plant cell walls’
Reply: Thank you for the constructive advice. We have revised the title as suggested.
Introduction
- Page 1, line 34. Sentence is difficult to follow. ….flexible enough to allow appropriate growth, on the other hand rigid enough to support….
Reply: Thank you for the advice. We have revised this sentence (line 33 to line 34).
- Page 1, line 39. Rewrite the sentence on chemical composition of primary and secondary cell wall. Give focus on similarities (cellulose, hemicellulose) and contrasts (pectin and lignin).
Reply: Thank you for the superb advice. We agree that this sentence should focus on the similarities and contrasts to make the meaning coherent of the preceding and following sentences. We have modified the sentence (line 38 to line 40).
- Page 1, line 43. Remove ‘dividing’
Reply: Thank you for pointing out this error, which was fixed in the revised text (line 43).
- Page 2, line 84. ‘embedded in pectic matrix’…
Reply: Thank you for this suggestion. This particular statement was modified in the revised manuscript (line 91).
- Page 2, line 87. Correct as ‘more recent’….
Reply: Thank you for pointing out this error, which was corrected in the revised text (line 94).
- Page 2, line 89. Give a brief description on monomer chemistry and organization as polymer with degree of polymerization of cellulose.
Reply: Thank you for the constructive suggestion. As suggested, we added a description on this question in the revised manuscript (lines 95 to line 99).
- Page 3, line 138. Remove ‘And’. Pectin has also been reported….
Reply: Thank you for pointing out this error and we have removed it in the revised text (line 149).
- Page 3, line 146. Correct as ‘significant change…’
Reply: Thank you for pointing out this error, which was fixed in the revised text (line 157).
- Page 4, Figure 1. The SEM figures of pollen grains. Are they from same plant or different? Staining used for cross section of stem. If Safranin is used, it should stain lignified cell wall as red and fast green stain non-lignified cell wall part as green. The provided figure looks like toluidine blue O staining. Please verify the method and provide correct information.
Reply: The SEM figures of pollen grains is from the different Arabidopsis plants. Thank you for pointing out this mistake. The staining used for cross section of stem is toluidine blue O staining. This particular statement was modified in the revised manuscript (line 75 to line 77).
- Page 5, line 224. AFM method in which mode ‘contact or non-contact mode during tapping?
Reply: The mode is peak force tapping. This particular statement was added in the revised manuscript (line 360).
- Page 5, line 226. Delete the repeating word ‘owing’
Reply: Thank you for pointing out this error, which was fixed in the revised text (line 243).
- Page 6, line 243. It is now possible to investigate ….
Reply: Thank you for pointing out this mistake, which was corrected in the revised text (line 265).
- Page 6, line 251. ..the wall by reducing mechanical stiffness but yet did not…
Reply: Thank you for this suggestion. This particular statement was modified in the revised manuscript (line 273).
- Page 6, line 274. Cellulose fibres or cellulose microfibrils?
Reply: Sorry for the misleading and this statement is cited from the reference (Daher et al., 2018, eLife). The original text is “anisotropic growth is common in plant organs and is canonically attributed to cell wall anisotropy produced by oriented cellulose fibers.”
- Page 8, line 348. Figure 2 does not provide immunolabelling figures. Adjust the ‘figure 2’ indication in the correct place. Perhaps as ‘AFM (Figure 2)
Reply: Thank you for the suggestion. We modified this description in the revised manuscript (lines 376).
Conclusion
- Page 9, line 361. Correct as ‘modification status’
Reply: Thank you for pointing out this error, which was fixed in the revised text (line 390).
Reviewer 3 Report
Comments and Suggestions for Authors
the work focusses on the significant part of the plant cell - its wall
however the presentation could be significantly improved
first the writing is poor- tense, word choice, omitted words, etc a sound scientific editing is essential for a quality paper
second there is a lack of figures how can you not show the different wall structures a composite figure and then details such as the variety of structures in pectins
third there is information that i would have added we now know that Ca flux in and out of cells ie into the cell wall is so important as a cell signaling event
we know too that the pectic fragments are active DAMPs ie these themselves have signaling properties
May be this was not the thrust of the paper but for me I do not see a clear focus in discussion - sections are on strength but then discussion goes into other directions
but there are no quantitative data on mechanical strength as detected by AFM a cartoon of how AFM works followed by images with values guiding the reader about the value of this technique would be expected
there is no discussion of pore sizes through the wall
or of an important role apoplastic water content
several comments as sticky notes at appropriate place not all suggested edits were marked though

poor word choice. tense changes ordering of information
ie needs good editing
Author Response
the work focusses on the significant part of the plant cell - its wall
however the presentation could be significantly improved
first the writing is poor- tense, word choice, omitted words, etc a sound scientific editing is essential for a quality paper
Reply: Thank you for the suggestion. We have sorted out the writing, adjusted paragraph order and corrected all the errors (line 179 to line 222 and all the revised words marked in red).
second there is a lack of figures how can you not show the different wall structures a composite figure and then details such as the variety of structures in pectins
Reply: Thank you for the advice. We have added another two figures to address your concerns. The different wall structures as shown in the revised figure 2 (line168 to line 172).
third there is information that i would have added we now know that Ca flux in and out of cells ie into the cell wall is so important as a cell signaling event
Reply: Thank you for the advice. In this review, we tried to provide a general understanding about the diverse structures and associated functional properties of plant cell walls. And there are already many comprehensive reviews (Alistair M. Hetherington and Brownlee, 2004; Tian et al., 2020; Dong et al., 2022; Cosgrove, 2024)
we know too that the pectic fragments are active DAMPs ie these themselves have signaling properties
May be this was not the thrust of the paper but for me I do not see a clear focus in discussion - sections are on strength but then discussion goes into other directions
Reply: Thank you for the advice. We modified the description in the revised manuscript (lines in red).
but there are no quantitative data on mechanical strength as detected by AFM a cartoon of how AFM works followed by images with values guiding the reader about the value of this technique would be expected
Reply: Thank you for the constructive suggestion. As suggested, we added another figure about how AFM works on this question in the revised manuscript (revised figure 3 in line 238 to line 241).
there is no discussion of pore sizes through the wall
or of an important role apoplastic water content
Reply: Thank you for the advice. We believe those content is not the focus of this review.
several comments as sticky notes at appropriate place not all suggested edits were marked though
Reply: Because the above comments lack punctuation and all sentences structure are incomplete, I cannot understand what you are referring as sticky notes.